# Decoherence in Excited Atoms by Low-Energy Scattering

**Diego A. Quiñones and Benjamin Varcoe ***

Experimental Quantum Information Lab, E. C. Stoner Building, University of Leeds, Leeds LS2 9JT, UK
pydaqo@leeds.ac.uk
* Correspondence: b.varcoe@leeds.ac.uk; Tel.: +44-113-343-8290

**Abstract:** We describe a new mechanism of decoherence in excited atoms as a result of thermal particles scattering by the atomic nucleus. It is based on the idea that a single scattering will produce a sudden displacement of the nucleus, which will be perceived by the electron in the atom as an instant shift in the electrostatic potential. This will leave the atom's wave-function partially projected into lower-energy states, which will lead to decoherence of the atomic state. The decoherence is calculated to increase with the excitation of the atom, making observation of the effect easier in Rydberg atoms. We estimate the order of the decoherence for photons and massive particles scattering, analyzing several commonly presented scenarios. Our scheme can be applied to the detection of weakly-interacting particles, like those which may be the constituents of Dark Matter, the interaction of which was calculated to have a more prominent effect that the background radiation.

**Keywords:** decoherence; Rydberg atoms; dark matter; axions

## 1. Introduction

Current experimental techniques have allowed the manipulation of atomic systems to previously unthinkable degrees, paving the way to the development of new technologies and the observation of very small quantum effects. One such technology is the quantum computer; trapped-ion systems have been implemented successfully to perform logical operations [1–3], making atomic systems a strong candidate for scalable quantum bits. Furthermore, the implementation of highly excited states (Rydberg atoms) has been proposed for quantum computing because of the length of their interaction and their long coherence times [3–6]. In order for atoms to be suitable for quantum technologies, it is necessary for them to have long coherence times, which is ultimately limited by their interaction with environmental particles [7]. In this paper, we propose a new decoherence mechanism of excited atomic systems induced by particles through short-distance interactions. This not only presents a fundamental limit to the stability of atomic systems, but can also be applied to the detection of weakly-interacting particles by means of analyzing the evolution of the atomic state. A search for exotic interactions using the decoherence of a Ramsey interferometer has been suggested previously [8,9], but the mechanism for linking the decoherence of internal degrees of freedom of the atom (energy levels) with momentum transfer was left open.

In our work, we propose that when a particle is scattered by the atomic nucleus it will produce an almost instant change in the position of the nucleus, which will be perceived by the electron in the atom as a sudden change in the electrostatic potential, projecting the wave function of the atom into the eigenstates of the new potential. First, we calculate the change in the state of the atom as a result of the nuclear displacement, obtaining a projection over lower energy levels only. We then express the order of the displacement in terms of the properties of the nucleus and the scattered particle, observing that the effect is more prominent for Rydberg atoms. Finally, we extend the analysis to

multiple scattering events and analyze the evolution of the state of an atom scattering photons and massive particles.

## 2. Decoherence by Scattering

A particle scattered by a nucleus will transfer some of its energy, causing a displacement of said nucleus. We suggest that if the translation of the nucleus is done in a very short period, the electron of the atom would not be able to follow the movement smoothly, resulting in a change of the internal state of the atom. The perturbation is presumed to be small enough so that no ionization will occur; any higher energy transference will not be accounted for by our model. As a result of the nuclear displacement, the electron will experience a sudden change in the electromagnetic potential, so its wave function will be projected into the eigenstates of the "new position" of the atom, in accordance with the adiabatic theorem [10].

Let $\psi_i(\vec{r})$ represent an eigenstate of the atomic system and $\psi(\vec{r} + \vec{r}_d)$ represent the state of the atom after the nucleus has been displaced along the vector $\vec{r}_d$. The state of the atom after the scattering can be expressed as a superposition of eigenstates as

$$\psi(\vec{r} + \vec{r}_d) = \sum_i C_i \psi_i(\vec{r}), \tag{1}$$

where the coefficients $C_i$ of the spectral decomposition can be calculated as

$$C_i = \int \psi_i(\vec{r}) \psi(\vec{r} + \vec{r}_d) d\vec{r}. \tag{2}$$

This equation can be very difficult to solve, depending on the atomic system. To simplify the calculations, we take some considerations and approximations that are described below.

First, we only consider initial states of the atom with no angular momentum. For atoms with just one electron in its last orbital—which are commonly used for atomic trapping [11–13]—we can approximate the wave function as that of the hydrogen atom by adding a correction term to the principal quantum number [14]; this approximation is especially accurate for Rydberg atoms [15].

Due to the stochastic nature of the particle scattering, it is not possible to know the direction of the displacement. Because of this, rather than consider a displacement along the vector $\vec{r}_d$, we will consider a delocalisation of the nucleus within the radius $r_d$. With these considerations, the wave function after the collision will be given by

$$\psi(r + r_d) = \frac{-1}{\sqrt{2}n_0 \cdot n_0!} \left( \frac{2}{n_0 a_0} \right)^{\frac{3}{2}} e^{-\frac{r+r_d}{n_0 a_0}} L^1_{n_0-1} \left( 2\frac{r + r_d}{n_0 a_0} \right), \tag{3}$$

where $L^1_{n_0-1}$ is the associated Laguerre polynomial, $a_0$ is the Bohr radius, and $n_0$ is the principal quantum number of the atom before the collision. The approximation in Equation (3) assumes the wave-function preserving a spherical symmetry after the collision, which restricts the interactions to those in which there is no exchange of angular momentum between the particle and the nucleus.

By algebraic manipulation of Equation (3), we get the expression

$$
\begin{aligned}
\psi(r + r_d) \\
= \frac{k_0^{\frac{3}{2}} e^{-\frac{k_0}{2} r_d}}{\sqrt{2} n_0 \cdot n_0!} \sum_{m=1}^{n_0} L^{-1}_{n_0-m} (k_0 r_d) \, e^{-\frac{k_0}{2} r} L^1_{m-1} (k_0 r) , \\
= e^{-\frac{k_0}{2} r_d} \sum_{m=1}^{n_0} \frac{m \cdot m!}{n_0 \cdot n_0!} L^{-1}_{n_0-m} (k_0 r_d) \, \psi_m(r) ,
\end{aligned}
\tag{4}
$$

with $k_0 = 2/a_0 n_0$. By using this identity in Equation (2), and since the eigenstates $\psi_i$ are orthogonal,

the coefficients of the decomposition can then be calculated as

$$
\begin{aligned}
C_n &= e^{-\frac{k_0}{2}r_d} \sum_{m=1}^{n_0} \frac{m \cdot m!}{n_0 \cdot n_0!} L_{n_0-m}^{-1}(k_0 r_d)\, \delta_{m,n}\,,\\
&= \begin{cases} e^{-\frac{k_0}{2}r_d} \frac{n \cdot n!}{n_0 \cdot n_0!} L_{n_0-n}^{-1}(k_0 r_d) & \text{for } n \le n_0 \\ 0 & \text{for } n > n_0\,. \end{cases}
\end{aligned}
\tag{5}
$$

This expression tells us that there is no excitation of the atom, but rather a projection over the initial state ($n = n_0$) and lower energy levels ($n < n_0$).

In Figure 1, the coefficients of the decomposition are plotted for different values of the displacement distance, illustrating the distribution of the wave function after the interaction. Here it is shown that the coefficients of lower-energy states increase with the displacement of the nucleus, which is expected because the effect should be more prominent with higher perturbation. It can also be appreciated that the coefficients decay very rapidly as their associated principal quantum number draws away from the quantum number of the initial state.

Next, we calculated the order of the perturbation as a function of the proprieties of the scattered particle.

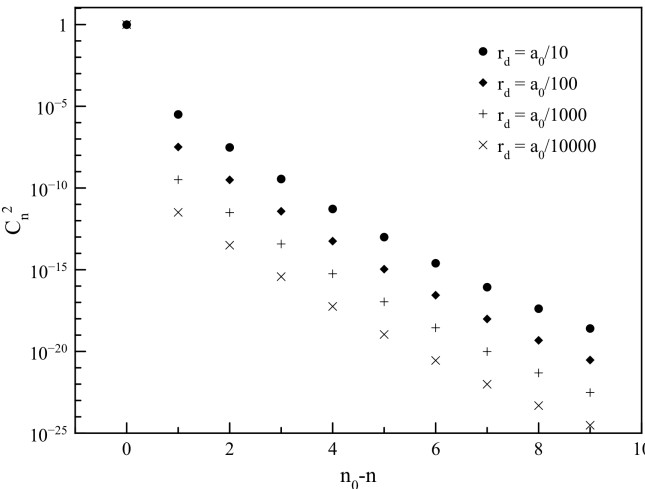

**Figure 1.** Coefficients of the eigenstate decomposition of the atomic state after the scattering of a particle by the nucleus (Equation (1)) for different values of nuclear displacement. The initial state of the atom has no angular momentum, and principal quantum number $n_0 = 10$.

## 3. Function of the Perturbation

The term $r_d$ is determined by how much the nucleus moves before the electron perceives its displacement. We estimated this quantity by taking the product of the velocity $\Delta v$ that the nucleus gains by the scattering multiplied by a time $\tau$; this period should be small enough to consider the change in the potential as non-adiabatic. For this time, we considered the period of the production of photons which carry the electromagnetic force between the nucleus and the electron, given by the equation

$$
\tau = \frac{\hbar^3}{4\mu^3}\left(\frac{n_0 m_e}{k_c e^2}\right)^2,
\tag{6}
$$

where $\mu$ is the reduced mass of the atom, $m_e$ is the mass of the electron, $k_c$ is the Coulomb's constant, and $e$ is the elementary charge constant. It takes this time for the nucleus to "send information" to the

electron about its position. The velocity of the nucleus is given by the energy $\Delta E$ it gains as a result of the scattering,

$$\Delta v = \sqrt{\frac{2\Delta E}{m_N}}, \tag{7}$$

where $m_N$ is the mass of the nucleus. Using Equations (6) and (7), we get the displacement distance

$$r_d = \Delta v \cdot \tau = \left(\frac{\hbar}{\sqrt{2}\mu}\right)^3 \left(\frac{n_0 m_e}{k_c e^2}\right)^2 \left(\frac{\Delta E}{m_N}\right)^{1/2}. \tag{8}$$

To give an idea of the order of the displacement, a cold neutron (kinetic energy of 0.025 eV) scattered by the atom will result in a displacement distance of $r_d/a_0 \sim 0.1$, according to Equation (8); other interactions presented in this paper will result in displacements of the nucleus several orders of magnitude smaller. Using Equation (8), we obtained that the coefficient of the initial state $C_{n_0}$ after the scattering (Equation (5)), will be given by

$$C_{n_0} = e^{-\frac{k_0}{2}r_d} = exp\left[\frac{n_0}{a_0}\left(\frac{\Delta E}{m_N}\right)^{1/2}\left(\frac{m_e}{k_c e^2}\right)^2\left(\frac{\hbar}{\sqrt{2}\mu}\right)^3\right]. \tag{9}$$

Using Equation (9), we generate the plot in Figure 2 of the value of $|C_{n_0}|^2$ as a function of $n_0$ for different energies transmitted to the nucleus; all used energies are below the ionization energy of the atom.

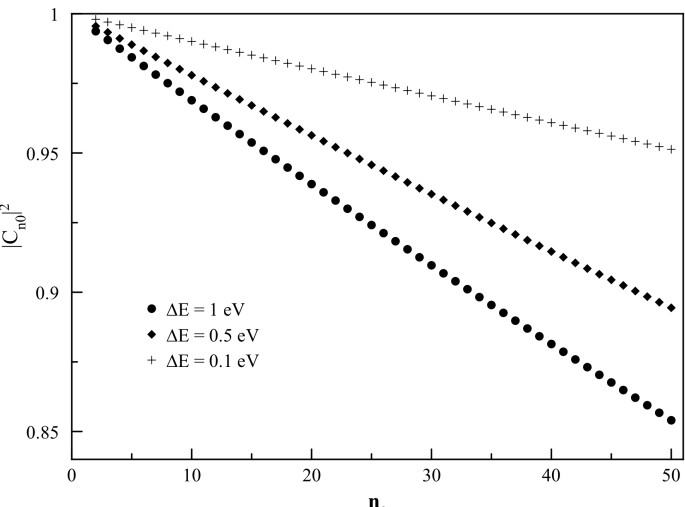

**Figure 2.** Probability of finding the atom in the initial state after a particle scattering by the nucleus as a function of the principal quantum number for different transmitted energies. The used mass for nucleus was $m_N = 1.66 \times 10^{-27}$ kg, which is approximately the mass of a Rubidium-87 nucleus.

It is observed that the magnitude of $C_{n_0}$ decays with the value of $n_0$, meaning that it will be more probable to find the atom in a state other than the initial if we prepare the atom in a highly excited state (Rydberg state). This result is reasonable, because states with high energy are more susceptible to perturbations by collision [16,17]. For our model, it can be argued that atoms in a high energy level have more possible states to decay to, and that the states have closer energy, which increase the probability of transition as a result of the interaction.

If we take interactions with angular momentum exchange into account, this would increase the possible decay channels, further decreasing the value of $C_{n_0}$. By performing the same calculations as

in Section 2 for a non-spherical symmetric wave function ($r \rightarrow r + r_d \cos\theta$), it can be obtained that the probability of transition to a state with $\Delta l = 1$ will be $\frac{2}{3}(r_d/n_0 a_0)^2$, which is similar to the transition to a state with $\Delta n = 1$ given in Equation (5); the change in the initial state for this case will be several orders of magnitude bigger than the calculated in Equation (9), so the studied model will correspond to the lower bound of the possible observable effect.

## 4. Temporal Evolution

It can expected that the atom will scatter multiple particles as time passes, so we calculate the coefficients of the decomposition after a given number of interactions. An atom with an initial state $\psi^{(0)}(r)$ will evolve into the state $\psi^{(1)}(r) = \psi^{(0)}(r + r_d)$ after a single scattering. When a second scattering occurs, the atom will change into the state $\psi^{(2)}(r) = \psi^{(1)}(r + r_d)$, assuming an interaction of the same order as for the first scattering and that the state of the atom does not evolve between collisions; any time dependence between consecutive collisions will increase the overall decoherence. The eigenstate decomposition of the state $\psi^{(2)}$ is calculated following Equations (1)–(5), resulting in

$$\psi^{(2)}(r) = e^{-2\frac{k_0}{2}r_d} \sum_{n'=1}^{n_0} L_{n_0-n'}^{-1}(k_0 r_d)$$
$$\times \sum_{n''=1}^{n'} \frac{n'' \cdot n''!}{n_0 \cdot n_0!} L_{n'-n''}^{-1}(k_0 r_d)\, \psi(r)_{n''} .$$

(10)

Here we have that the coefficients are given by

$$C_n^{(2)} = e^{-2\frac{k_0}{2}r_d} \frac{n \cdot n!}{n_0 \cdot n_0!} \sum_{n'=n}^{n_0} L_{n_0-n'}^{-1}(k_0 r_d)\, L_{n'-n}^{-1}(k_0 r_d) .$$

(11)

We then calculate the coefficients for the state after a third scattering $\psi^{(3)}(r) = \psi^{(2)}(r + rd)$, using the same logic as for the state $\psi^{(2)}(r)$, and so on until we calculate the coefficients for an arbitrary number of interactions. The resulting general formula that gives the coefficients after a given number of particle scatterings is too long and complicated to be shown in this paper. As we have previously indicated, the coefficients decay very rapidly as its principal quantum number goes farther from the number $n_0$ (Figure 2), so we focus our attention on the coefficient of the initial state $C_{n_0}$ and the eigenstate one energy level lower $C_{n_0-1}$; after $l$ number of collisions, these coefficients are given by

$$C_{n_0}^{(l)} = e^{-l\frac{k_0}{2}r_d},$$

(12)

$$C_{n_0-1}^{(l)} = -l(k_0 rd)\frac{n_0-1}{n_0^2}e^{-l\frac{k_0}{2}r_d} .$$

(13)

The above calculations assume that the collisions are statistically independent. This approximation is accurate for our model because the probability of interaction is expected to be very small, so enough time will pass between two collisions to neglect any correlation [16]. Additionally, it will be seen below that very long times are necessary to observe the proposed effect, further validating the approximation.

It is important to call attention to Equation (13); here we have that the coefficient $C_{n_0-1}$ is not an exponential function of the number $l$ (which is time-dependent, as shown next). An exponential dependency of the time is typical for the effect of other sources of decoherence, like energy relaxation of the electron [18]. This difference will allow the effect to be distinguishable over other mechanisms of decoherence. Another important remark is that, because of the dependency of $n_0$, the change in population in the atom's energy levels can be modulated by using a different initial state, which is useful for experimental observation.

To estimate the number of collisions over time, we consider only short distance interactions between the particles and the nucleus. The number of collisions in a period $t$ is given by

$$l(t) = \sigma F_p t , \tag{14}$$

where $\sigma$ is the effective cross-section, $F_p$ is the flux of particles, and $t$ is the time. By substituting this expressions in Equation (12), we obtain the time evolution of the coefficient $C_{n_0}$.

$$C_{n_0}(t) = e^{-\frac{k_0}{2} r_d \sigma F_p t} . \tag{15}$$

We proceed to estimate the magnitude of the decoherence, first for the scattering of photons and then for the scattering of particles with mass.

### 4.1. Photon Scattering

An alternative form of Equation (16) that is useful to calculate the number scattered of photons is

$$l(t) = \sigma \frac{\eta_E c}{h\nu} t , \tag{16}$$

where $\eta_E$ is the energy density of the electromagnetic radiation, $c$ is the speed of light, and $\nu$ is the light's frequency.

For low energy photons, we consider Thomson scattering by the atomic nucleus, which gives the cross-section

$$\sigma_T = \frac{8\pi}{3} \left( \frac{k_c e^2}{m_N c^2} \right)^2 . \tag{17}$$

The same formula can be applied for photon scattering by the electrons of the atom; because of their small mass, we will disregard this interaction in our calculations. The average change in the energy of the photon, given in the rest frame of the nucleus, is

$$\Delta E = \frac{h^2 \nu^2}{m_N c^2} \tag{18}$$

Using Equations (15), (17), and (18), we finally arrive to the function

$$C_{n_0}(t) = exp \left[ -\frac{\sqrt{8}\pi n_0 \eta_E m_e^2 \hbar^3}{3 a_0 \mu^3 m_N^3 c^4} t \right] . \tag{19}$$

With this formula, we calculate how the state of an atom differs from its original state as a function of time for different energy densities of radiation surrounding the atom (Figure 3).

The analyzed energy densities correspond to (a) the solar radiation on the Earth's surface, considering a constant insolation of 52.2 PW ($\eta_E = 8.49$ MeV/cm$^3$) [19], (b) the environment of an AMO laboratory with the ambient lights turned on, given a measurement of 4 μW with a detector of 9.5 mm in diameter ($\eta_E = 1.17$ KeV/cm$^3$), and (c) the cosmic microwave background ($\eta_E = 0.25$ eV/cm$^3$) [20]. These cases represent common circumstances for atoms to be exposed to, and cover a fair range of energy densities. The time scale of the plot comprise periods that are much larger than the radiative lifetimes of Rydberg atoms [21]. In experiments, it should not be expected to directly see the loss in population due to the proposed effect, but rather a deviation in the decay curves due to spontaneous emission and other known perturbation sources. The effect of the photon scattering is shown to be very small—even for the most energetic radiation—but its contribution can be relevant for ultra-precise metrology.

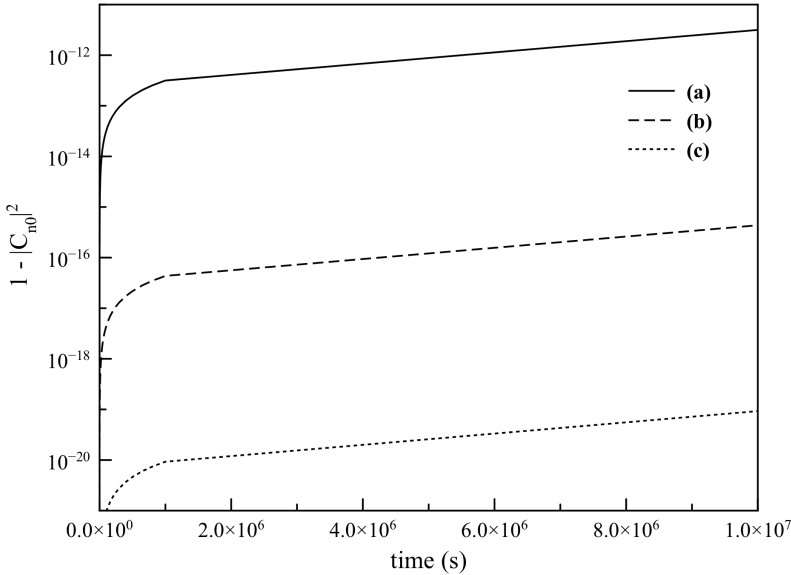

**Figure 3.** Difference in the population of the initial state of an atom with $n_0 = 60$ and $m_N = 1.66 \times 10^{-27}$ kg as a function of time. The plots correspond to the scattering of (**a**) solar radiation ($\eta_E = 8.49$ MeV/cm$^3$); (**b**) ambient lights in the laboratory ($\eta_E = 1.17$ KeV/cm$^3$); and (**c**) the cosmic microwave background ($\eta_E = 0.25$ eV/cm$^3$).

Other studies have analyzed the decoherence induced by the scattering of non-resonant photons, but they are based on the localization of the atom by the scattered light ([7,22]). In these studies, the interaction of stochastic radiation is ignored, which is covered by our model.

### 4.2. Massive Particle Scattering

For the scattering of a particle with mass, we consider the interaction as a direct collision between the particle and the nucleus. The energy transferred to the nucleus for a scattering event at the most probable angle is given by

$$\Delta E = \frac{\mu}{m_N m_e} \left( m_p v_p \right)^2 , \tag{20}$$

where $m_p$ and $v_p$ are the mass and the velocity of the scattered particle, respectively. With this formula we have that the coefficient of the initial state evolves as

$$C_{n_0}(t) = exp \left[ -\frac{n_0 \sigma F_p m_p v_p}{\sqrt{8} a_0 m_N} \left( \frac{\mu}{m_e} \right)^{1/2} \left( \frac{m_e}{k_c e^2} \right)^2 \left( \frac{\hbar}{\mu} \right)^3 t \right] , \tag{21}$$

We use this formula to calculate the state of an atom continuously scattering two different kinds of neutral particles (Figure 4).

We analyze two particular cases that we consider to be the most significant in terms of limiting the stability of atomic systems: (1) The scattering of neutrons, and (2) the scattering of Dark Matter. In case (1), we consider neutrons from secondary cosmic rays having a flux of $F = 2 \times 10^4$ neutrons per second per square meter, a cross-section of $\sigma = 3$ barn, and a kinetic energy of 0.07 GeV [23]. In case (2), we analyze the local distribution of Dark Matter with a reported density of $5.41 \times 10^{-22}$ kg/m$^3$ [24]. We assumed that Dark Matter is thermalized by the background radiation ($T = 2.726$ °K) and that it is composed of Axions, with a mass of $m_p = 1$ eV/c$^2$ and a cross-section of $\sigma = 0.01$ barn [25,26]. For experimental observation, all axion–electron interactions can be excluded, as they will result in excitation or even ionization of the atom [27,28].

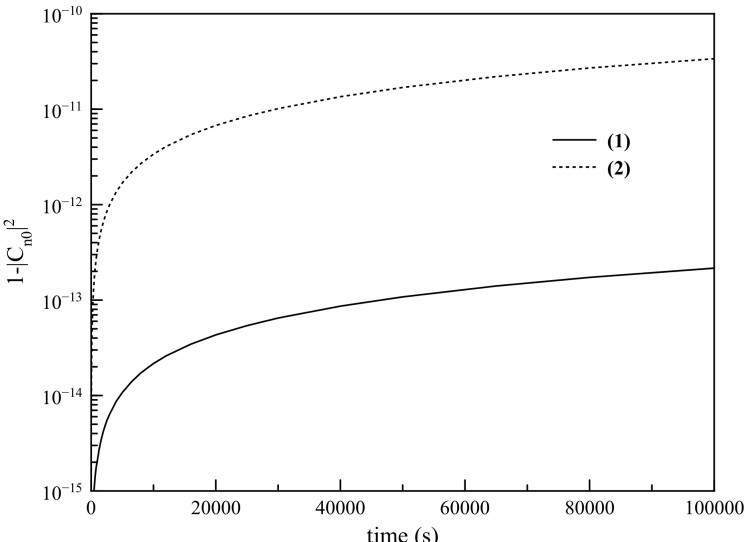

**Figure 4.** Difference in the initial state of an atom with $n_0 = 60$ and $m_N = 1.66 \times 10^{-27}$ kg as a function of time. The plots correspond to the interaction with (**1**) neutrons from secondary cosmic rays, and (**2**) local Dark Matter composed of Axions.

The decoherence by massive particle scattering is observed to be more prominent than for photon scattering in both cases. This means that the scattering of cosmic rays could impose the ultimate limit to the stability of excited atomic systems.

It is remarkable that the particular case of local Dark Matter we analyzed will perturb the atoms to a higher degree than the cosmic rays. This presents the possibility of applying our theory to the detection of such particles, or to set a limit to their mass, by analyzing the coherence in highly stable atomic systems, like the hydrogen maser [29,30]. The interaction of exotic matter has already been modeled as a s-wave scattering, with the proposed method of detection very similar to ours [26].

## 5. Conclusions

The scattering of a particle by the nucleus of an excited atom will result in the atom having its wave function partially projected into lower-energy states. The probability of finding the atom in a certain state decays exponentially as its associated principal quantum number differs from the one of the initial state. It was calculated that the coefficient of the initial state would be lower for atoms in a highly excited state, making the effect more prominent for Rydberg atoms. The decoherence produced by photon scattering was calculated to be very small, but significant enough to be relevant in ultra-precise or long-lasting experiments. For massive particle scattering, the resulting decoherence is considerably higher. Our scheme can be applied to the detection of neutral particles, or at least to impose a limit to their properties.

**Acknowledgments:** The author Diego A. Quiñones thanks the Mexican National Council for Science and Technology (CONACYT) for the granted financial support.

**Author Contributions:** The authors have contributed equally to the formulation of the theory and the calculations where performed by Diego A. Quiñones. Both authors have read and approved the final manuscript.

**Conflicts of Interest:** The authors declare no conflict of interest.

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
