# Peer review of "Decoherence in Excited Atoms by Low-Energy Scattering"

_atoms, doi:10.3390/atoms4040028_

Round 1
Reviewer 1 Report
Report of “Decoherence in Excited Atoms by Low-Energy Scattering
The manuscript (MS) studied the decoherence of excited atoms as a result of thermal particles scattering by the atomic nucleus. It is based on the idea that a single scattering will produce a sudden displacement of the nucleus, which will be perceived by the electron as an instant change in the electrostatic potential. The decoherence was shown to increase with the excitation of the atom. The MS estimated the order of the decoherence for photon and massive particle scattering. The MS discussed the detection of weakly-interacting particles, like those which may be the constituents of Dark Matter.
My comments of the MS are:
1. The decoherence of excited atoms is a result of thermal particles scattering by the atomic nucleus. Therefore, it is important to know the order of magnitude of a nuclear displacement caused by thermal particles. The MS did not estimate this magnitude.
2. A particle scattered by a nucleus will transfer some energy, causing a displacement of the nucleus with respect to its original rest frame. The nucleus gain some energy as a result of the scattering. The authors suggest that, if the translation of the nucleus is done almost instantaneously, the electron of the atom would not be able to follow the movement smoothly, resulting in a change of the internal state of the atom. Such kind of collision processes is inelastic. But the MS used the assumption of the elastic collision to estimate the average number of collisions per unit time and obtain Eq. (14). It is a question how the elastic collision is appropriate in the study of the MS.
3. The state of the atom is a function of position r, which is a vector, in Eq. (1). Then the state of the atom becomes a function of scalar r in Eq. (3). The wave function of the atom after the collision depends on the moving direction of the nuclei. Instead of│r+rd│, why the state of the atom after the collision can be a scalar r+rd ? If the wave function of the atom after the collision depends on│r+rd│, Eq. (4) of the MS is then invalid.
In conclusion, the MS in the present form is not suitable to be published unless the MS can discuss more physical mechanisms to above comments.

Author Response
We appreciate your comments and feedback. We have performed several changes to our manuscript, providing more relevant references and addressing your comments in the following way:
- We include a calculation of the displacement resulting from a single scattering that serves as an example of its order of magnitude. This calculation was done for a cold neutron scattering, having in mind experimental tests for our model.
- We described the scattering of the particle by the nucleus as elastic collision to indicate that there is no energy dissipation during their interaction.
If after the interaction the atom is measured in a lower energy state, then there was a transformation of binding energy of the electron into kinetic energy, which describes an inelastic collision as you correctly call attention to. In the new manuscript, we rephrased the description of the interaction.
- The scattering of thermal particles by the nucleus is a stochastic process. It is not possible to know the direction of the collision, and therefore the path of the displacement, so we rather take a radius of delocalization for the nucleus with respect to its original position for our calculations. We included this important explanation in the manuscript.
Reviewer 2 Report
In this manuscript, the authors calculate the electronic rearrangement in Rydberg atoms after the scattering of particles on the atomic nucleus. They find, that a sudden displacement of the nucleus results in contributions of lower-lying states in the Rydberg atom. Though this effect is very weak for the elastic scattering of photons (Thomson scattering) and neutrons, it is surprisingly strong for the scattering of Dark Matter particles (Axions).
I find the idea of Dark Matter scattering on Rydberg atoms very appealing as a theoretical system. However, I have some reservations on the practical relevance of the discussed phenomena for several reasons (see questions below). In my opinion, this manuscript is worth to be published but some question should be clarified and discussed a bit more in detail before:
- The effect discussed in this paper takes place on a time scales of days (Fig. 3 and 4). The typical radiative lifetimes of Rydberg atoms are usually many orders of magnitude smaller (e.g. Branden, et al., J. Phys. B 43, 015002 (2010)). That should be mentioned by the Authors.
- In the examples detailed in chapt. 4, the scattering of the particles on the electrons also contributes (and have even a larger cross section than the scattering on the nucleus, see e.g. Ref [15]). Also this point should be discussed somewhere in the manuscript.
- Page 7, line 136-139: I was not able to reproduce the estimate on the Axion scattering cross sections based on the information given in Ref [15]. In [15] the ionization of atoms is considered due to the interaction of electrons with rather high Axion energies which is a very different scenario than the one discussed in the present manuscript. Maybe, the authors can give some more details on their estimate.
There are some more formal comments:
- Page 6 line 103: It should read “The number of collisions in the period t is given by …”. Eq. 14 does not give the number of collisions per unit time.
- Lines 5 and 25: In my opinion, the term “projection” is not used in a proper way. A projection of a state is always connected to the “reduction” of the state to a sub space. This is not the case here, but it is rather a change of the basis. The projection would only take place by measuring the contribution of a specific n.
Author Response
We appreciate your comments and feedback. We feel that you present valid concerns that seek to produce a manuscript worth of publication.
We have performed several changes to our manuscript, providing more relevant references and addressing your comments in the following way:
- We provide a brief discussion about the Rydberg atoms’ lifetimes (including the references you kindly provided), explaining that the experimental observation of the effect should not be by direct observation, but rather analysing the deviation of the expected loss in population in big ensembles of atoms.
- For photon scattering, the cross section of the photon-electron interaction is much smaller compared to the photon-nucleus one (for Thomson scattering). For massive particle scattering, the interaction with the electrons will produce an excitation of the atom, which does not contribute to decoherence in our model as we only consider transitions on lower energy level. These arguments, along with proper references, are included in the new version of the manuscript.
- The reference provided was not the correct one. We provide now the appropriate reference in which the number we use is within the range of cross section proposed for the axion-nucleon interaction.
- We changed the text as suggested.
- We use the term “projection” in an algebraic sense. To prevent any further confusion, in the new manuscript we use the term only when talking about the mathematical transformation, avoiding it in chapters where we discuss experimental schemes.
Round 2
Reviewer 1 Report
Although the authors explained the stochastic nature of the particle scattering and considered the delocalisation of the nucleus within a radius. We still have to average the wavefunction of Eq. (3) over the scattering angle to obtain the radial component of the wavefunction. We cannot just replace the position vector by the scalar quantities. The final form of Eq. (4) depends on the averaged wavefunction of Eq. (3).
Author Response
The constant r_d does not represent necessarily the magnitude of the vector of displacement, but rather is the delocalization radius, as stated just before equation 3. Any constant resulting on the averaging over all scattering angles is included in this coefficient. When we calculate the magnitude of r_d (equations 8, 18 and 20) we take into account all possible scattering angles and the distribution of energy for the colliding particle.
Reviewer 2 Report
In the revised manuscript, the authors addressed some of the referees’ questions appropriately. However, there is at least one important question which should be discussed in more detail:
· I agree with the concerns of the other referee regarding the replacement of the displacement vector rd by its absolute value. This seems to over-simplify the physical process for at least three reasons:
- Considering the information on the displacement direction, the wave function will not be spherically symmetric (with respect to the new position of the nucleus) after the collision. Therefore, there must be contributions of non-zero orbital angular momenta. Just neglecting that seems to be a very rough estimate.
- For two or more scattering processes, neglecting the direction of rd corresponds to a ‘loss of memory’ of the atom, because it does not ‘remember’ where it was coming from. That corresponds implicitly to an incoherence between two scattering events. It is not clear to me, if that incoherence is a reasonable assumption and if equation (10) and (11) are a proper averaging over all directions.
- Already after the first collision, the wave function is a superposition of non-degenerate states, i.e. it is a time-dependent wave packet. The used approximation does not account for any time dependence between two collisions.
A more detailed discussion on the shortcomings and limits of the approximation should be added going beyond just stating that the direction is not known.
And again a minor point:
· I still find the (algebraic) use of the term ‘projection’ inaccurate in the manuscript. Just an example: The phrase following equation (5) states that there is a '... projection over the initial state and lower energy levels'. I guess, the authors want to say that there is a superposition of the initial state and lower energy levels.’ Though I think, an accurate language is extremely important, I consider that being a very minor point.
I want to add some comments on the form of the revised document: It would greatly facilitate the work of the referee, if the authors would clearly indicate which paragraphs that have been modified in the revision process. It is very tedious to compare the two versions without knowing where it has been changed. Moreover, the line numbering got inconsistent in the revised version.
Author Response
Thank you for your feedback. I would like to respond to your comments, and explain how we addressed the points you made in the version of the manuscript we are currently submitting:
- We focused our calculations to interactions where there is no exchange of angular momentum between the particle and the atom. By allowing the opposite case, more decay channels are accessible which will increase the order of decoherence in the atomic state (by several orders of magnitude, as we calculated). Therefore, our model corresponds to the lower bound of the possible observable effect, i.e. the minimum decoherence experienced by the atom as a result of the particle scattering. This discussion is added in page 3, line 6 and page 5, line 5 of the new version of the manuscript.
- Because in our model the atom undergoes a dynamics similar of that of a particle in a vicious thermal medium, it is logical to assume that it will be properly modelled as a non-Markovian random process, as you point out. In our model, the low probability of the interaction (specially for massive particle scattering), will result in very long times between consecutive collisions, reducing the statistical correlation between them. Furthermore, the experimental observation of the effect will require very long periods of observation and a big number of atoms in the ensemble, which will average out the fluctuations. We explain this in page 5, line 15 of the new version.
- A time-dependent dynamics of the system between collisions will add a random phase to the system, effectible increasing the decoherence of the system. Our consideration of no evolution will then give again the lower bound of the expected decoherence. This argument is included in page 6, line 9.
- We change the text to avoid further use of the term “projection”. This can be seen in page 3, line 16.
We haven’t figure out how to correct the line numbering, so please pay not attention to it. Each of the stated changes in the manuscript corresponds to the line within the mentioned page; we hope this will make the modification easier to find.
Round 3
Reviewer 1 Report
The manuscript satisfies the rigorous criteria of a paper in the present form. I recommend to publish the manuscript now.
Reviewer 2 Report
The questions have been addressed appropriately.
Author Response
We will like to take the opportunity to thank you for your feedback which help us to improve the content of the manuscript.